# The Biological Activity of Tea Tree Oil and Hemp Seed Oil

**Marietta Lakatos, Samuel Obeng Apori, Julie Dunne and Furong Tian ***

School of Food Science Environmental Health, Technological University Dublin, City Campus, Grangegorman, D07ADY7 Dublin, Ireland; maricsuli07@gmail.com (M.L.); samuel.obengapori@tudublin.ie (S.O.A.); julie.dunne@tudublin.ie (J.D.)

**\*** Correspondence: furong.tian@tudublin.ie; Tel.: +353-899607414

**Abstract:** The interest in hemp seed oil (HSO) and tea tree oil (TTO) in the medical and food industries is increasing. The current study compares their bioactivity to other plant oils, mainly focusing on hemp seed oils (HSOs) with various cannabidiol (CBD) contents. A DPPH assay was employed to evaluate the antioxidant activity. The antimicrobial activity against *Escherichia coli*, *Staphylococcus aureus*, and *Salmonella enteritidis* was evaluated using time–kill, minimum inhibition concentration (MIC), and Kirby–Bauer disk diffusion methods. Tea tree oil showed significantly higher antimicrobial activity against *S. enteritidis* compared to *E. coli* and *S. aureus* ($p < 0.05$). The antioxitant activity range (lowest to highest) was sesame < vetiver < rosehip < tea tree < organic hemp < pure hemp < 5% CBD < vitamin C. Tea tree oil and 5% CBD showed antioxidant activity at IC50 of 64.45 μg/mL and 11.21 μg/mL, respectively. The opposing antimicrobial and antioxidant results for TTO and HSO indicate that these activities arise from different components within the oil compositions.

**Keywords:** bioactivities; antimicrobial properties; tea tree oil; hemp seed oil; CBD oil; antioxidant properties; plant extracts

## 1. Introduction

Plant extracts have been widely investigated for their bioactive properties. There has been an increased demand from consumers for more natural methods to enhance food safety and quality. Due to this, as well as increased environmental concerns, there is an emphasis on substituting synthetic antioxidants with plant extracts. Due to their antimicrobial and antioxidant properties, bioactive hemp seed oil (HSO) [1] and tea tree oil (TTO) are particularly of interest in medical and research fields.

Tea tree oil is an essential oil from the plant Melaleuca alternifolia. Its common name is tea tree or narrow-leaf paperbark. It has been used for its antimicrobial properties for decades in Australia, where the plant originated [2]. Tea tree is a shrub or tree with papery bark that can grow up to 14 m and thrives in warm temperatures. It is a member of the Melaleuca genus and the myrtle family [3]. The common method for plant extraction is steam distillation. The active components of tea tree oil are terpenes, mainly monoterpenes [2]. Tea tree oil is mainly used topically as an antiseptic, and it can be purchased in health shops without a prescription. It can be purchased as an essential oil, but it is also an ingredient in many cosmetics. For example, it is a popular ingredient in products marketed to prevent or treat acne [4,5].

While hemp production has a long history, recently the hemp derived from Cannabis Sativa has received interest as an ingredient in the food industry [4]. After harvesting, hemp seeds are usually cold-pressed to extract the hemp seed oil (HSO) [5]. HSO can be produced to be low or high in cannabidiol (CBD) content. High-CBD products can also differ from each other. Both hemp tincture and CBD oil are hemp products high in CBD, and their names are sometimes confused [6]. To date, there has been no comparison of the bioactivity of TTO and HSOs with different CBD contents.

While plant oils are well-known for their antibacterial effects, many also exhibit antifungal properties. There is an ongoing need to improve the antimicrobials available for use in the food industry due to resistance to antimicrobials, as well as the possible adverse effects of harsh chemicals used for sanitizing food-processing areas. Consequently, finding antimicrobial agents that are naturally occurring and that have no toxic effects is an important area of research [7].

Antioxidants are molecules that are produced naturally or synthetically to neutralize harmful reactive oxygen species (ROS). They are essential to avoid oxidative stress, which arises when there is shift in the balance between oxidants and antioxidants in favor of oxidants. This can be caused by external factors, such as smoking and alcohol consumption or exposure to the sun without sun protection. When this imbalance occurs over extended periods, it can have a toxic effect on the body. Elevated ROS levels are associated with aging and various chronic diseases, such as diabetes, cancer, and cardiovascular disease [8]. The body has systems in place to combat normal levels of oxidants; however, severe excessive levels can have detrimental health effects.

Plant extracts containing antioxidants are often also found to be effective antimicrobials. Together, these two characteristics make a material very valuable for scientific study [9,10]. Sesame oil is a popular cooking oil in Asian cuisine, while rosehip seed oil is an ingredient often used in cosmetics. Vetiver oil is an essential oil made from the grass *Vetiveria zizanioide*, which is grown in tropical climates. It was found to increase alertness in an in vivo study [11]. Sesame oil and vetiver oil are employed as control groups in this study.

*Salmonella* is one of the leading bacterial food pathogens that can cause illness. There are many strains of Salmonella, the most commonly found—and the strain used in this research—is *Salmonella enteritidis* (*S. enteritidis*) [12,13]. The bacteria *Staphylococcus aureus* (*S. aureus*) and *Escherichia coli* (*E. coli*) are common bacteria causing mild symptoms [14,15]. The current study compares HSO and TTO products and investigates their antioxidant activity using a DPPH assay and antimicrobial activity against *S. aureus*, *E. coli*, and *S. enteritidis*. HSO and TTO are studied separately as antioxidants and antimicrobials. The study also investigates any interrelationships between the antioxidant and antimicrobial properties.

## 2. Materials and Methods

### 2.1. Materials

2,2-diphenyl-1-picrylhydrazyl was sourced from Sigma Aldrich (Burlington, MA, USA), methanol from Fisher Scientific (Waltham, MA, USA), and TSA agar and Mueller–Hinton agar were purchased from Sigma-Aldrich. Pure vitamin C powder was purchased from Holland and Barrett (Nuneaton, UK). The oils used were Grano Vita organic hemp seed oil, Jacob Hooy 5% CBD oil, Pure Hemp tincture, Xpel tea tree oil, Rainbow Abby vetiver oil, Miaflora 100% organic rosehip seed oil, and Ottogi sesame oil.

Ultrapure deionized water (resistivity > 18.0 MΩ cm$^{-1}$) was used for all the solution preparations and experiments. *Escherichia coli* ATCC 25922, *Staphylococcus aureus* ATCC 25923, and *Salmonella enteritidis* ATCC 13076 were cultured.

### 2.2. Antimicrobial Assay

For these experiments, isolated colonies of bacteria were required. A complex streak was performed on TSA plates using the original bacteria. Once the streak was complete, the petri dishes were incubated at 35 °C for 24 h. The protocol also recommended the bacteria be used the next day, as it was important to ensure growth was still in the log phase. First, the inoculum was prepared: sterile loops were used to pick up 1–2 colonies of *Escherichia coli* (*E. coli*), *Staphylococcus aureus* (*S. aureus*), and *Salmonella enteritidis* (*S. enteritidis*) from the plates.

Time–kill studies were carried out as described previously [16]. Briefly, bacteria cells were diluted in fresh medium, grown to the exponential phase, and then diluted again in medium to adjust cell densities to approximately $10^4$ CFU/mL. Seven different oil types

were added at concentrations of 1 mg/mL of the MIC. Rates of killing were determined by measuring the reduction in viable bacteria (log10 CFU/mL) at 0, 1, 2, 4, 6, and 24 h at fixed concentrations of the compounds. Experiments were performed in duplicate. If plates contained fewer than 10 CFU/mL, the number of colonies was considered below the quantitation limit. Samples of the culture-containing compounds were diluted 10-fold to minimize oil carryover to the CAMHB plates. The antimicrobial activities detected using the MIC of seven oils were evaluated against three bacteria using broth dilution method [16]. Seriel dilutions of tea tree oil ranging from 16 to 0.015 mg/mL were prepared in 100 mL volumes with bacteria in a 96-well microtiter tray for 24 h.

The standardized version of the Kirby–Bauer disk diffusion protocol was followed, as described by Hudzicki [17]. The bacteria were spread on a Mueller–Hinton Agar plate, and paper discs containing the antimicrobial agent were also placed on the media. The antibacterial activity was quantified by a clear zone of inhibition around the indicator zone of the disc. This method is preferred when investigating the antimicrobial properties of plant extracts due to its low cost and ease of implementation. Furthermore, despite newly developed technologies, the disk diffusion assay is still considered reliable when screening for antimicrobials [18]. The current study performed the Kirby–Bauer method using three strains of bacteria and seven oil samples. The antimicrobial disks were divided into three petri dishes for the experiment. Experiments were carried out in duplicate.

Mueller–Hinton agar plates were placed on a bench, and a sterile swab was inserted into the inoculum. The excess liquid was removed by rotating the swab on the side of the tube. Following this, the plate was streaked three times, each time turning the plate by 60 °C to create a lawn of bacteria. Afterwards, the swabs and the loops were safely discarded, while the plates were allowed to dry for 10 min. During this time, the plates were divided. The watch glasses were wiped and dried through evaporation near a Bunsen flame using ethanol for sterilization. The plastic tweezers were also sterilized using ethanol in a beaker. A 0.1 mL aliquot of each oil was placed on the watch glass, and the paper discs were dipped in the oils. After the uptake of the oil, the excess was removed. Using the tweezers, the discs were placed on the agar plates. The plates were then incubated at 37 °C for 48 h. This was performed in duplicate for all the bacteria evaluated. After 48 h, any changes were recorded, and the inhibition zone was measured.

### 2.3. Antioxidant Assay

When evaluating the concentration of antioxidants in a sample, the extraction technique, experimental conditions, and the type of assay chosen each need to be considered [19]. Several methods can be used to evaluate antioxidant activity. One of the earliest and most-utilized is the 2,2-diphenyl-1-picryl-hydrazyl-hydrate (DPPH) radical-scavenging activity assay. DPPH is a stable, free radical; the assay works by measuring the scavenging activity of the antioxidant. Due to its deep violet color, the absorbance of this compound is best measured at the wavelength of 517 nm. DPPH is often used because it is suitable for hydrophobic and hydrophilic antioxidants; it is inexpensive to perform and is designed to allow weak antioxidants to be evaluated. The limitations include that DPPH can only be dissolved in nonpolar solvents, which may interfere with the absorbance values during quantitative analysis. DPPH is also light-sensitive [20].

Vitamin C is considered an excellent antioxidant and is widely used as a positive control for relative comparison in DPPH assays [21]. The DPPH method was applied to several oil products, and the total antioxidant capacity was compared to vitamin C.

The vitamin C, as a positive control, was diluted to different concentrations (30, 25, 20, 15, 10, and 5 µg/mL). The concentrations of the oil samples were adjusted in methanol (0.1 µg/mL to 100 µg/mL). To carry out the DPPH inhibition assay, a 1.0 mL aliquot of each sample (vitamin C or oil samples at various concentrations) was added to a test-tube, after which 3.0 mL of 40 ppm DPPH solution was added. The test tubes were left in the dark for 30 min, after which the contents of the test tubes were transferred to a plastic cuvette. Methanol at 99.9%was used as the reference. A Shimadzu RF-6000 spectrameter

was used to measure the absorbance at 520 nm. The DPPH radical-scavenging activity was calculated using the following formula:

$$\text{DPPH scavenging effect (\% inhibition)} = \{(A0 - A1)/A0\}{*}100\}$$

where A0 is the absorbance of the control reaction, and A1 is the absorbance in the presence of all of the samples and the blank. The 50% inhibition (IC50) was calculated from a graph plotting the inhibition percentage against the oil concentration. All of the oil sample IC50 values were compared to the IC50 value of vitamin C. Tests were carried out in triplicate.

### 2.4. Statistic Analysis

ANOVA was employed to evaluate the differences between the samples and the controls using the SPSS 11 statistical analysis package. Statistical significance was set to $p < 0.05$. A one-way analysis of variance (ANOVA) was used to test the differences among the groups of oils. Exploratory data analyses were performed with Kolmogorov–Smirnov tests to validate the normality assumption found in the ANOVA and t-tests. A robust graphical method for testing the equality of variances [22], similar to the analysis of means (ANOM), was used to identify those groups that were significantly different ($p < 0.5$) and accounted for the variance demonstrated.

## 3. Results

### 3.1. Antimicrobial Assay

For time–kill curves for the bacterial pathogens, compounds were added to the cultures at time zero, and the samples were processed as described in Section 2. The bacterial strains used were *S. enteritidis, E. coli*, and *S. aureus.*

The onset of bacteria growth was at 6 h, and the highest concentration was observed at 24 h (Figure 1). The numbers of bacteria in the TTO group were lower than the other groups, with satistical differences at incubation times of 4 h, 6 h, and 24 h ($p < 0.05$).

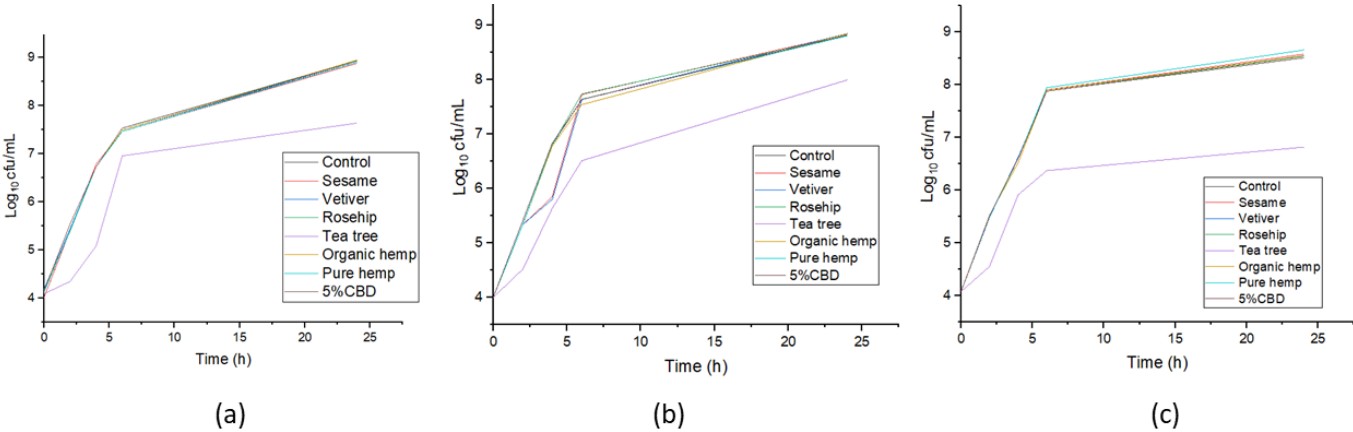

**Figure 1.** The time–kill curves of seven oils against (**a**) *E. coli*, (**b**) *S. aureus* and (**c**) *S. enteritidis*. The untreated control, sesame, vetiver, rosehip, organic hemp, pure hemp, and 5% CBD lines overlapped with each other. The purple lines indicate the tea tea oil antimicrobial activity.

The MIC and disk diffusion method tested the antimicrobial activities of seven oils on *S. Aureus, E. coli*, and *S. enteritidis*. The experiments were repeated three times in duplicate. However, only TTO exhibited antimicrobial activity among all the groups. The measurement of the inhibition zone diameter is indicated in Figure 2a,b, whilst Table 1 shows the MIC and average inhibition zones of TTO against the different bacteria.

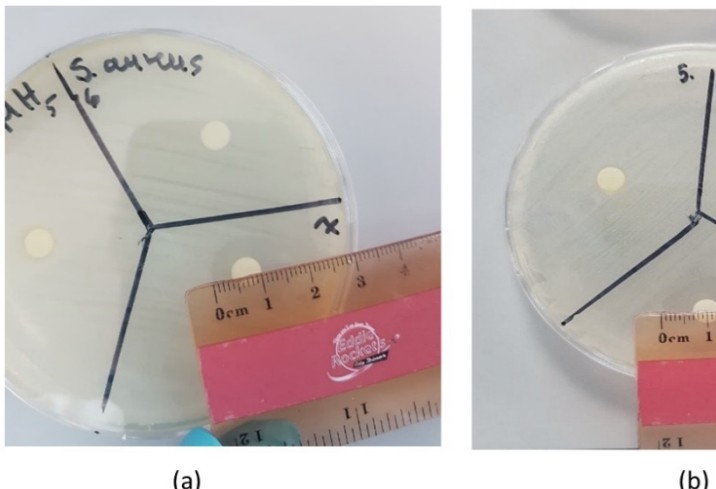

(a) (b)

**Figure 2.** The measurement of the inhibition zone of tea tree oil on different bacteria: (**a**) *S. aureus* and (**b**) *E. coli*.

**Table 1.** The inhibition zones of TTO against the different microbes.

| Tea Tree Oil on Different Bacteria | MIC (mg/mL) | Average Diameter of Inhibition Zone (cm) |
|:---:|:---:|:---:|
| *S. aureus* | 8 | $1.87 \pm 0.12$ |
| *E. coli* | 8 | $2.02 \pm 0.15$ |
| *S. enteriditis* | 2 | $2.19 \pm 017$ * |

* $p < 0.05$ compared to the groups of *S. aureus* and *E. coli*.

The MIC ranges of the tea tree oil were 8, 8, and 2 mg/mL for *S. aureus*, *E. coli*, and *S. enteritidis*, respectively. The average inhibition zones were 2.19 cm, 2.02 cm, and 1.87 cm against *S. enteritidis*, *E. coli*, and *S. aureus*, respectively. The tea tree oil showed significant antimicrobial activity. There was no antimicrobial activity from the control groups (Rainbow Abby vetiver oil, Miaflora 100% organic rosehip seed oil, and Ottogi sesame oil), HSO, or CBD oil. There was a statistical difference between *S. enteritidis* compared to *E. coli* and *S. aureus* for tea tree oil ($p < 0.05$).

### 3.2. Antioxidant Assay

A series of vitamin C solutions at various concentrations was prepared and analyzed using a DPPH assay, and an image of these is shown in Figure 3.

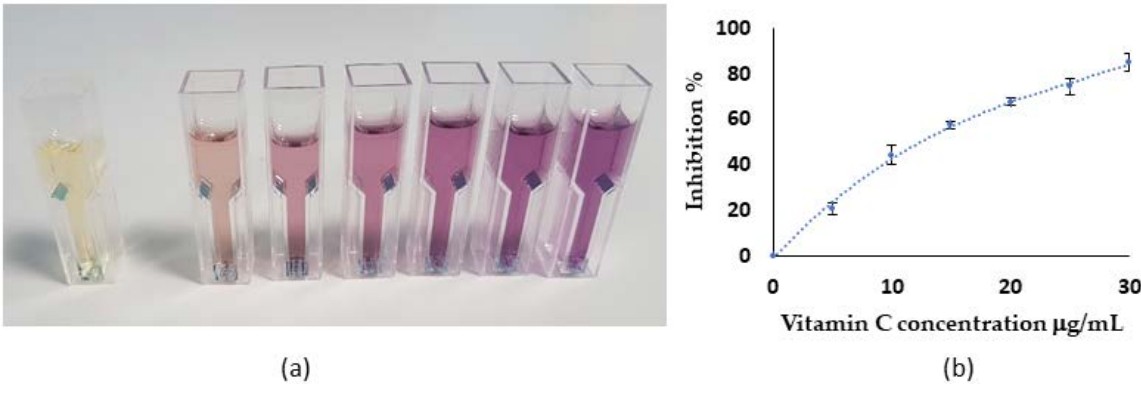

(a) (b)

**Figure 3.** DPPH assay of different concentrations of vitamin C. (**a**) Image of DPPH assay with vitamin C in cuvettes. From left to right: stock solution, 30, 25, 20, 15, 10, and 5 μg/mL. (**b**) The vitamin C DPPH scavenging percentage of inhibition ($n = 3$), showing IC50 of 10 μg/mL.

Figure 3a shows the assay colors corresponding to various concentrations of vitamin C antioxidants, confirming the successful chemical reaction and the expected results. The vitamin C reacted with DPPH to instantaneously produce a yellow color, while the diluted samples displayed a milder color change. The DPPH scavenging effect (percentage of inhibition) vs. the vitamin C concentration (triplicate assays) is shown in Figure 3b. Vitamin C exhibited free-radical-scavenging properties with an IC50 of 10 μg/mL.

After carrying out the DPPH assay with the oil samples, absorbance was used to calculate the IC50 values. The graph below demonstrates the IC50 values of the antioxidant activities of the different oils (Figure 4).

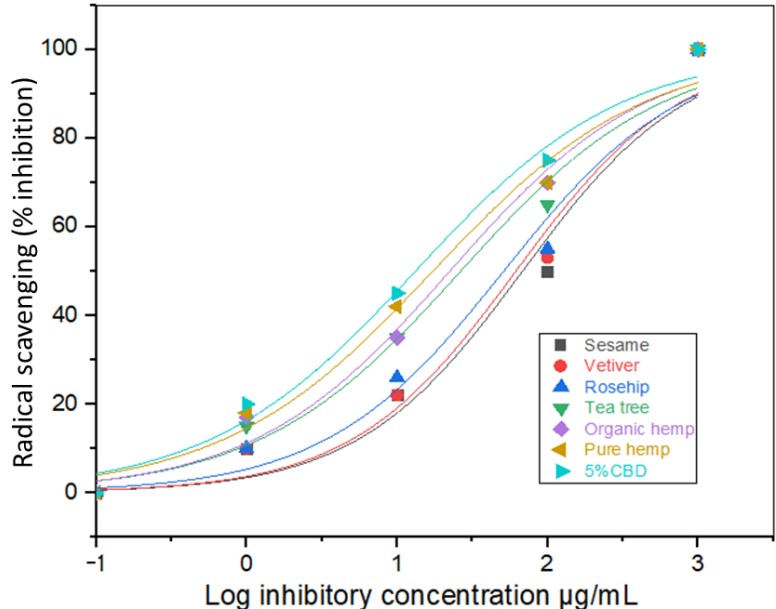

**Figure 4.** IC50 values of the antioxidant activities of the different oils.

The 5% CBD oil produced the highest antioxidant activity (Table 2 and Figure 4). The 5% CBD hemp seed oil performed better than other oils. It was found to be approximately 5.7 times more potent than TTO. There was a statistically significant difference between the 5% CBD oil and the other groups ($p < 0.05$).

**Table 2.** The IC50 values of antioxidant contents of the oil samples and vitamin C.

| Sample | IC50 (μg/mL) |
|---|---|
| Sesame | 81.81 ± 9.08 |
| Vetiver | 80.21 ± 8.10 |
| Rosehip | 78.89 ± 10.1 |
| Tea tree | 64.45 ± 5.11 |
| Organic hemp | 20.35 ± 2.10 |
| Pure hemp | 17.56 ± 0.98 |
| 5% CBD | 11.21 ± 0.21 * |
| Vitamin C | 10.00 ± 0.18 |

* $p < 0.05$ compared to the other oil groups.

The IC50 values showed the antioxidant activity as follows: sesame < vetiver < rosehip < tea tree < organic hemp < pure hemp < 5% CBD < vitamin C. The IC50 values of 5% CBD, pure hemp, and organic hemp were recorded at 11.21 μg/mL, 17.56 μg/mL, and 20.35 μg/mL, respectively. This proved that CBD exhibits promising antioxidant properties. It can also be seen in Figure 4 that each HSO presented a different antioxidant potential. The numbers were determined based on the amount of CBD reported in the product. The reported CBD contents in organic hemp, pure hemp and 5% CBD oil were 0.2 mg/mL,

20 mg/mL, and 54 mg/mL, respectively. According to these findings, the concentration of CBD played a significant role in antioxidant activity, with higher CBD content related to higher antioxidant activity in the oil samples. There was a significant statistical difference between 5% CBD oil, organic hemp, and pure hemp ($p < 0.05$).

## 4. Discussion

Tea tree oil has been shown to possess antimicrobial activity by different scientific methods, and some studies also suggested that hemp seed oil has antimicrobial effects [23]. The current study aimed to confirm these findings and to compare the effectiveness of the oils' antimicrobial properties.

The tea tree oil data obtained in the present study are similar to that from previous studies against *E. coli, S. aureus*, and *C. albicans*, which found MIC ranges of 2–24 mg/mL [16]. Tea tree oil showed significant antimicrobial activity against *S. enteritidis* compared to *E. coli* and *S. aureus*. In contrast, none of the three hemp seed oils (Grano Vita organic hemp seed oil, Jacob Hooy 5% CBD oil, and pure hemp tincture) presented any antimicrobial activity in the current study. In addition, it is important to note that the different HSOs had different CBD contents; however, the differences did not affect the antimicrobial activity [20]. Therefore, the current study results are consistent with the literature. The antimicrobial activity exhibited by TTO is caused by monoterpenes inhibiting bacterial respiration. Cox et al. concluded that the antimicrobial activities of TTO were due to microbial susceptibility to monoterpene-induced cell membrane damage [24], while Brun et al. also found that TTO showed good antimicrobial activity [25].

In the current study, *S. enteriditis* plates displayed the greatest inhibition zones, followed by *E. coli* and *S. aureus* (Figures 1 and 2 and Table 1), which is in agreement with reported studies [24,25]. Additionally, Brun et al. found that TTO products exhibited inhibition against antibiotic-resistant *S. aureus* and *Pseudomonas aeruginosa* [25]. The antimicrobial effect exhibited by TTO may be assigned to terpinen-4-ol, which is the primary constituent of TTO [26,27]. The terpinen-4-ol constituent causes cell death by irreversibly destroying the microbial membrane and inducing the loss of cytoplasm, as reported previously [28]. Furthermore, the antifungal activity of TTO was demonstrated by Mondello et al. [29], who found that TTO was effective in vitro and in vivo against vaginal infections caused by *C. albicans.*

However, the current study contradicts reported findings regarding the antimicrobial activity of HSO, given that the only positive result obtained was with TTO. A previous study reported that hemp seed oil displayed antimicrobial activity against *Staphylococcus* and *E. coli* [18]. However, our group's recent work found that the extraction method used to obtain the HSO could greatly influence the antimicrobial activity [1]. Different methods were compared in that study, which found that solvent extraction with methanol and cold pressing produced good results. However, hydrodistillation produced less microbial inhibition [1]. Therefore, the ineffectiveness of HSO against *S. aureus, E. coli*, and *S. enteriditis* may be due to the method used to extract the HSO [28]. It was also reported that wild hemp may have better antimicrobial potential compared to registered cultivars. Zheljazkov et al. used oils that were extracted using hydrodistillation [30]. According to Jadhav et al., oils extracted with hydrodistillation may be superior as antimicrobial agents [31].

Although TTO is known for its antimicrobial properties, few studies have reported its antioxidant effects. An in vitro study by Zhang et al. investigated the antioxidant activity of TTO with DPPH, thiobarbituric acid reactive species (TBARS), and a hydroxyl radical-scavenging method [32]. Our results are consistent with their work, with TTO showing lower antioxidant activity than vitamin C (Table 2). In the current study, the antioxidant activity was not only compared to vitamin C, but also to other essential oils. This current study is the first to report that TTO showed lower antioxidant activity than HSO, but higher activity than sesame, vetiver and rosehip oils (Table 2 and Figure 4). In the case of bioactive HSO, antioxidant activity arises from its tocopherols content [33]. It was reported that HSO could reduce oxidative stress in vivo in *D. melanogaster*, suggesting that HSO could act

as an antioxidant on the cellular scale through high levels of polyunsaturated fatty acids and tocopherols [34]. Another study in vitro reported that HSO could be a good source of antioxidant intake through diet [35]. The current study suggested that CBD content was directly related to antioxidant activity, as the product with the lowest IC50 inhibition (Figure 4). These observations agree with another recent study. Kitamura et al. found that six of seven hemp seed oil products contained CBD. The two oils specifically marketed as CBD presented better antioxidant activity [36]. Furthermore, Citti et al. identified tetrahydrocannabinol and cannabidiol, as well as 30 other cannabinoids, for the first time in hemp seed oil using liquid chromatography-mass spectrometry [37], with significant variances in the cannabinoid contents of the HSO samples. Further study must be conducted to investigate the relationship between CBD and antioxidant activity.

Previous studies have suggested there is a relationship between the antimicrobial and antioxidant effects in HSO and TTO [1,32]. The current results showed that antioxidant and antimicrobial activity were not linked in the cases of TTO and HSO. Brophy et al. investigated 800 TTO samples for their compositions and discovered components that were not reported before, as well as variances between the samples. The same study also described that, during the storage period, some of the terpene constituents, such as α-terpinene and γ-terpinene, oxidized and converted to p-cymene [5]. These findings suggested that the compositional changes in an essential oil may affect its antimicrobial and antioxidant activity. More research should be conducted to uncover and optimize the antimicrobial and antioxidant activity of this oil.

## 5. Conclusions

It was shown that, while HSOs with higher CBD contents provided better antioxidant activity, TTO was the most effective antimicrobial agent against *S. enteritidis, E. coli*, and *S. aureus*.

This study is the first to report that tea tree oil showed significantly better antimicrobial activity against *S. enteritidis* compared to *E. coli* and *S. aureus*. Tea tree oil showed modest antioxidant activity with an IC50 of 64.45 μg/mL. The 5% CBD showed the highest antioxidant activity at an IC50 of 11.21 μg/mL, with the higher CBD content likely making it the best antioxidant. Both plants are valuable, with the potential to be used in functional products. Whether in the food industry or in agriculture, they can be helpful in food science studies against microbes and ROS that can make food products unsafe for consumers. Finding ways to use these plants and to optimize their effects is an important area of research. The variations caused by CBD content should be further investigated, as this is an important aspect of the oil. Although TTO showed modest antioxidant power, it may be beneficial to investigate TTO samples from different producers to compare the impact of the source on the this activity.

**Author Contributions:** Conceptualization, M.L. and F.T.; methodology, M.L.; validation, M.L., S.O.A. and F.T.; formal analysis, M.L.; investigation, J.D.; resources, J.D.; data curation, M.L.; writing—original draft preparation, M.L.; writing—review and editing, S.O.A. and J.D.; supervision, F.T.; project administration, J.D.; funding acquisition, J.D. All authors have read and agreed to the published version of the manuscript.

**Funding:** This research received no external funding.

**Institutional Review Board Statement:** Not applicable.

**Informed Consent Statement:** Not applicable.

**Data Availability Statement:** Not applicable.

**Acknowledgments:** We thank Gouri Atapattu for her technical support and early text reviewing.

**Conflicts of Interest:** The authors declare no conflict of interest.

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
