# Peer review of "The Biological Activity of Tea Tree Oil and Hemp Seed Oil"

_2673-8007, doi:10.3390/applmicrobiol2030041_

Round 1

Reviewer 1 Report

The authors describe the biological activity of tea tree oil and hemp seed oil. 

The authors suggest that the present study aimed to confirm the activities of these oils and compare the effectiveness of their antimicrobial properties.

However, I don't think there is much merit compared to previous papers. As the authors described, there have been many previous papers regarding the bioactivities of these oils.

Also, the antioxidant effects of the oils suggested by the authors are not very impressive.

Author Response

Dear Reviewer:

Thanks for your comments. We have taken this time to improve our manuscript. The new concepts of TTO and HSO have been emphasized in abstract, introduction, material and method, results and discussion. The responses are in blue in the attached file and revised version of manuscript.

Reviewer 2 Report

The work by Tian et al. describes a comparison of the biological activity of tea tree oil, hemp seed oil, and other oils. The manuscript is well written. The research design and methods are good. In general, the quality of the study is good with modest bioactivity.

Minor corrections.

1. Change "ml" to "mL" throughout the manuscript. 

2. Explain a little bit more about using Vitamin C in the DPPH Assay other than just adding reference 17. 

Author Response

Thanks for your comments. We have taken this time to improve our manuscript. The responses are in red in response and text in revised manuscript.

Author Response

Thanks for your comments. We have taken this time to improve our manuscript. The new concepts of TTO and HSO have been emphasized in abstract, introduction, material and method, results and discussion. The English language and grammar have been proved by native English speaker author. The responses are in green in attached file and revised manuscript.

Round 2

Reviewer 1 Report

I think this manuscript has been improved over the previous version.

Author Response

Dear reviewer,

Thanks for taking your time to review the manuscript and your comments. We have seen the gaps in research design and references styles. All reference style has been checked and mended.
If we are using IC 50 of DPPH assay, we do not need Vitamin C as standard. Sorry for confusion.
The vitamin C has been removed as standard from previous text, because all of the oil sample have IC50 value. 
The vitamin C show better radical scavenging inhibition as a positive control. The IC 50 of Vitamin C has been included in the table 2. It is also included in the result, discussion and abstract.
The corrections are in blue colour through the text.

Furong

Reviewer 3 Report

Please, see the file attached.

Author Response

Dear reviewer,
Thanks for taking your time to review the manuscript and your comments. We have seen the gaps in research design and result presentation. We accepted your corrections in red colour. English language and style spell have been checked.

We clarified the experiment design that the IC 50 of Vitamin C has been included as a positive control. It is also included in the result, discussion and abstract.
The corrections are in blue colour through the text.

Furong
